# Prevalence, associated factors, and association of intimate partner violence and suicidal behaviors among women of reproductive age in Asia: Protocol for a systematic review and meta-analysis of cross-sectional studies

**Mantaka Rahman**[ID][1,2*], **Habiba Kabir**[3], **Anika Naowar Chowdhury**[2], **Md. Babu Raihan Mia**[2], **Tamal Saha**[1], **Afroza Tamanna Shimu**[4], **Ummul Khair Alam**[2]

**1** International Centre for Diarrheal Disease Research, Bangladesh (icddr,b), Dhaka, Bangladesh, **2** National Institute of Preventive and Social Medicine (NIPSOM), Dhaka, Bangladesh, **3** Bangladesh Medical University (BMU), Dhaka, Bangladesh **4** Dhaka Medical College and Hospital, Dhaka, Bangladesh

* drmantaka.icddrb@gmail.com

## Abstract

Intimate partner violence (IPV) is strongly linked to suicidal behaviors (ideation, plans, attempts), affecting 1 in 3 women globally, with prevalence varying across regions. Sociocultural and economic factors shape IPV and SB risk differently. This review will estimate prevalence, risk factors, and associations of IPV and SBs among women of reproductive age in Asia to provide region-specific evidence for targeted interventions. We will systematically search PubMed, Scopus, PsycINFO, Web of Science, EMBASE, CINAHL, and Google Scholar for studies published up to 30 November 2025, following PRISMA guidelines. The search will also include grey literature and citation chaining, using keyword truncation, string searches, and standardized indexing terms. Cross-sectional observational studies among Asian women (19–45 years) exposed to intimate partner violence (physical, psychological, sexual) will be included, reporting suicidal behaviors (ideation, plans, attempts) compared with unexposed women. Only English-language, peer-reviewed studies will be considered, while reviews, abstracts, and unpublished studies will be excluded. Two independent reviewers will screen studies for the central concepts of "Intimate Partner Violence (IPV)" and "Suicidal Behaviors (SB)", with disagreements resolved by a third reviewer. Data on prevalence, associated factors, mediator variables, and numerical estimates of IPV–SB associations will be extracted. Meta-analysis using a random-effects model will be conducted alongside a narrative synthesis. Findings will be visualized with forest and funnel plots, heterogeneity assessed using the Q Cochrane statistic and I² index, and subgroup and sensitivity analyses performed. Risk of bias will be evaluated using the Joanna Briggs Institute (JBI) Critical Appraisal Checklist. Early identification of psychological distress using culturally validated tools, combined with understanding context-specific drivers of IPV, is essential for

**Data availability statement:** All study related data including search strings, and relevant data is attached with the manuscript as a supplementary file. Data will be made available publicly on Kaggle when the study will be completed and published.

**Funding:** The author(s) received no specific funding for this work.

**Competing interests:** The authors have declared that no competing interests exist.

**Abbreviations:** IPV: Intimate Partner Violance; SB: Suicidal Behaviors; PRISMA: Preferred Reporting Items for Systematic Reviews and Meta-Analyses; MOOSE: Meta-Analysis of Observational Studies in Epidemiology.

preventing suicidal behaviors among Asian women of reproductive age. Findings from this review will inform targeted interventions, guide policy, and address gender norms that perpetuate violence and elevate mental health risks.

## Introduction

Global evidence indicates a strong link between intimate partner violence (IPV) and suicidal behavior (SBs), including suicidal ideation (SI), plans (SP), and attempts (SA). Suicidal behaviors comprising thoughts, plans, and non-fatal self-directed actions with the intent to die is a major contributor to morbidity and mortality worldwide [1]. IPV, a common form of interpersonal violence and a significant public health concern, affects nearly one in three women globally over their lifetime [2]. IPV and suicide mortality often co-occur globally; therefore, individuals presenting with suicidal distress or following self-harm should be routinely assessed for IPV [3]. A recent English study reported that 49.7% of past-year suicide attempters had ever experienced IPV, with past-year IPV at 23.1%, both linked to higher odds of self-harm (OR 2.20) and suicidal thoughts (OR 1.85) [3]. Most notably, half of the global burden of female suicides is concentrated within the 11 WHO South-East Asia Region countries, driven by sociocultural factors, societal norms, stigma, and male-dominated structures [4]. Despite the exceptionally high burden of suicide morbidity and mortality in the WHO South-East Asia Region, evidence on its risk factors is limited, and underreporting likely leads to underestimation of the true figures [1, 4].Global evidences have documented regional variation in women's IPV and suicidal behaviors: Asia (IPV 21–35%, SB attempts 4–5%, ideation 14%) [2], Europe (IPV 16–23%, SB attempts 11–12%, ideation 13–16%) [5], Americas (IPV 25%, SB 6.7–8%), Australia (IPV ~ 23%, SB attempts 5–6%), and Africa (IPV 40–44%, SB attempts 11–12%, ideation >20%) [5, 6] (Fig 1). Additionally, another concerning finding is that nearly 56% of female suicides and 40% of male suicides occur in individuals under 30 years of age particularly in low-and-middle income countries (LMICs) [4]. A recent meta-analysis reported a moderate, direct association between IPV (primarily physical and sexual) and suicidal outcomes, with IPV survivors having 3–5 times higher odds of suicidality [7].

The triggering factors for IPV and suicidality vary across countries, influenced by sociocultural, economic, and legal contexts. While a study from Asian countries identified post-conflict settings, shifting gender roles, economic stress, and psychosocial factors as key drivers of IPV and suicidal behavior among women [8], in Europe psychological (IPV mostly abuse experience and economic) IPV have been strongly linked to suicidal ideation [9], and in sub-Saharan Africa, controlling behavior is deeply rooted in poverty, alcohol use, and patriarchal norms and is associated with both IPV and increased mental health risk including suicidality [10]. The cultural, interpersonal, and practical barriers are one of the major hurdles to care seek for IPV survivors causing psychological distress

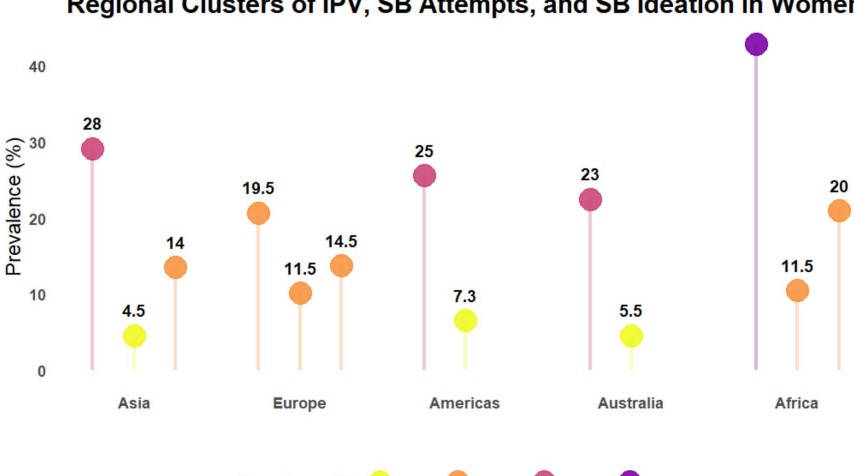

**Fig 1. Regional clusters of IPV, SB (suicidal attempts, and suicidal ideation) among women.** The illustrative background was adapted from prior literature to display regional prevalence across Asia, Europe, the Americas, Australia, and Africa, using a color gradient ranging from 0% to 50% to represent varying prevalence levels across regions.

and further prawn to SBs. Addressing root risk factors like economic stress, social norms, and trauma through evidence-based IPV prevention interventions could reduce IPV by about 15% or more, which may in turn lower SB risk [11].

Despite growing evidence linking IPV to adverse mental health outcomes, the specific contribution of IPV to SBs including as ideation, planning, and attempts among women of reproductive age (15–49 years) remains unclear [8]. Existing reviews have often examined IPV broadly in relation to multiple mental health outcomes, without isolating suicidal behaviors or differentiating the impacts of physical, sexual, and psychological IPV. As a result, opportunities to identify and target a key modifiable risk factor for female suicidality had been missed. Furthermore, a recent meta-analysis [7, 12] had estimated the association of IPV and SBs while the present systematic review protocol is completely apart from the reporting by focusing specifically on women of reproductive age in Asia, estimating the prevalence of IPV, and suicidal behaviors (SI, SA, SP), and identifying association, and associated risk factors, using only cross-sectional studies to provide region-specific, comparable evidences. This focused approach will provide actionable documentation to inform targeted, culturally adaptive interventions and policies for suicide prevention in Asian women.

## Materials and methods

### Study objectives and research question

The primary objective of this study will be to investigate the association between IPV and suicidal behaviors (suicidal ideation, suicidal plan, and suicidal attempts) among Asian women of reproductive age groups. The secondary objective will be to find the pooled prevalence of IPV and SB across the group in Asia and explore the possible associated factors. There will be two research questions: (1) What is the pooled prevalence of IPV and SB among women aged 15–49 years in Asia? (2) What is the association between exposure to IPV and suicidal ideation, suicidal plan, and suicidal attempts among women of reproductive age in Asia?

## Study methods

**Design and registration.** The study protocol has been prospectively registered in International Prospective Register of Systematic Reviews (PROSPERO), [13] under the unique registration number CRD420251238459 (URL: https://www.crd.york.ac.uk/PROSPERO/recorddashboard).This review will adhere to the Preferred Reporting Items for Systematic Reviews and Meta-Analyses (PRISMA 2015) [14] and Meta-Analysis of Observational Studies in Epidemiology (MOOSE) [15] guidelines.

## Search strategy

A comprehensive search strategy, incorporating Medical Subject Headings (MeSH) terms will be adopted to extract relevant literature from specified electronic databases, and citation chaining of grey literature. A number of databases, including PubMed (MEDLINE), Scopus (Elsevier), PsycINFO (APA PsycNet), Web of Science (Clarivate Analytics), EMBASE (Elsevier), CINAHL (EBSCOhost), and Google Scholar (Google), will be accessed up until November 30, 2025. The search strategy will include two central concepts of "Intimate Partner Violence (IPV) " and "Suicidal Behaviors (SB)" among Asian women. The searched keywords, detailed screening process, and PEO criteria (Participants, Exposure, and Outcome) are depicted in PRISMA 2020 Flowchart [16] (Fig 2). Only studies published in English-language in internationally peer-reviewed journals reporting observational cross-sectional studies will be included. In addition, women who have not experienced domestic violence, will also serve as a comparison group to identify differences with the women who were victimized by IPV.

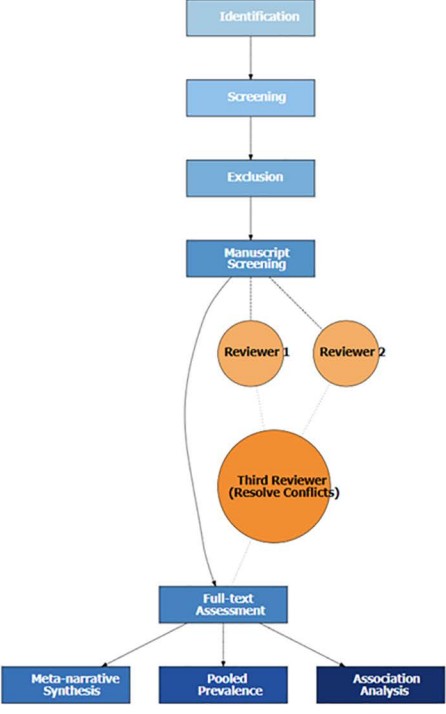

**Steps of PRISMA Flowchart for study identification, screening, assessment, and inclusion process**

**Fig 2. PRISMA flow diagram for systematic review and meta-analysis.** The flowchart illustrates the study identification, screening, assessment, and inclusion process at each stage, with full-text articles subsequently considered for meta-analysis, narrative synthesis, and association assessment.

## Eligibility criteria

Based on the PEO components, the eligibility criteria for the review will be as such:

1. **P (Population):** Women (19–45 years) of Asian countries who have experienced domestic violence perpetrated by an intimate partner, representing different socio-cultural contexts and varying degrees of abuse.

2. **E (Exposure):** The multifaceted impacts of domestic violence, including both direct and indirect effects arising from emotional, physical, psychological, and sexual abuse.

3. **O (Outcome):** The elevated risk of suicidal behaviours, encompassing SI, SP, and SA, among women who have experienced domestic violence compared with those unexposed.

## Study selection process

The primary (title and abstracts), and secondary (full-text) screening of all retrieved studies will be assessed by two independent reviewers based on the predefined inclusion and exclusion criteria. The relevant studies will be finally included for meta-analysis. Any disagreements regarding the eligibility of a study will be discussed and resolved through consensus.

## Outcomes (primary and secondary)

The primary and secondary outcome of the study will be to assess the strength of association and pooled prevalence of IPV and SB including SI, SA, and SP respectively among women in Asia. In addition, the study will also include exploration of heterogeneity and potential sources of variation. as well as, identification of associated factors, and moderator variables influencing the association between IPV and SB Table 1.

## Methodological quality assessment

The methodological quality of the finally included studies will be evaluated using the Joanna Briggs Institute (JBI) critical appraisal checklists for Analytical Cross-Sectional Studies. Only cross-sectional studies will be included, as this design will provide direct and comparable estimates of prevalence and associated factors at a defined time point, ensuring methodological consistency and reducing heterogeneity across the included studies. The JBI checklist comprises eight items covering key domains such as sampling adequacy, measurement reliability, confounding control, and statistical analysis. Each item will be rated as "Yes" (1 point), "No" (0 point), "Unclear", or "Not applicable." A total score will be calculated for each study selection, with a maximum possible score of 8. Based on the overall score, studies will be classified as follows: High quality (low RoB): 6–8 points, Moderate quality (moderate RoB): 4–5 points, low quality (high RoB): ≤ 3 points. Two reviewers will independently perform the appraisal, and disagreements will be evaluated through consensus or by consulting a third reviewer. The influence of methodological quality on pooled estimates and effect sizes will be explored through subgroup or sensitivity analysis.

## Study selection and data management

The EndNote™ 21.0 reference management software (Clarivate Analytics, Philadelphia, USA) will be used to organize all articles retrieved from the database searches. Search results will be imported into EndNote when necessary, where duplicate records will be identified and removed. Additionally, any studies selected through manual or reference list searches (citation chaining) will be added to the EndNote library as needed. After de-duplication, the remaining records will be exported to Rayyan QCRI, a web-based systematic review management platform, to facilitate independent screening of titles and abstracts by two reviewers and to ensure efficient collaboration throughout the study selection and review process.

**Table 1. Example search strategy for pubmed database (CoCoPoP Framework).**

| Search Element | Search Terms and Structure |
|---|---|
| **Condition (Co)** | ("Intimate Partner Violence*"[MeSH] OR "Domestic Violence"[MeSH] OR "Spouse Abuse"[MeSH] OR "intimate partner violence"[Title/Abstract] OR "domestic violence"[Title/Abstract] OR "spousal abuse"[Title/Abstract] OR "partner violence"[Title/Abstract] OR "partner abuse"[Title/Abstract] OR "spouse abuse"[Title/Abstract] OR "wife beating"[Title/Abstract] OR "battered woman"[Title/Abstract]) |
| | ("Suicide"[MeSH] OR "Suicidal behaviour*"[MeSH] OR "Suicide, Attempted"[MeSH] OR "Self-Injurious Behavior"[MeSH] OR "suicidal ideation"[Title/Abstract] OR "suicide ideation"[Title/Abstract] OR "suicide attempt"[Title/Abstract] OR "attempted suicide"[Title/Abstract] OR "self-harm"[Title/Abstract] OR "self-harm"[Title/Abstract] OR "self-injury"[Title/Abstract] OR "self injur*"[Title/Abstract]) |
| | ("risk factors"[Title/Abstract] OR "associated factors"[Title/Abstract] OR "determinants"[Title/Abstract] OR "predictors"[Title/Abstract] OR "correlates"[Title/Abstract] OR "contributors"[Title/Abstract] OR "protective factors"[Title/Abstract] OR "modifiable factors"[Title/Abstract] OR "non-modifiable factors"[Title/Abstract]) |
| **Population (P)** | ("Female"[MeSH] OR female*[Title/Abstract] OR woman [Title/Abstract] OR women [Title/Abstract] OR "women of reproductive age"[Title/Abstract] OR "reproductive-age women"[Title/Abstract] OR "reproductive age"[Title/Abstract] OR "15–49"[Title/Abstract]) |
| **Context (Co)** | ("Asia*"[MeSH] OR Asia[Title/Abstract] OR "South Asia*"[Title/Abstract] OR "Southeast Asia*"[Title/Abstract] OR "East Asia*"[Title/Abstract] OR "Western Asia*"[Title/Abstract] OR "Central Asia*"[Title/Abstract] OR Afghanistan[Title/Abstract] OR Bangladesh[Title/Abstract] OR Bhutan[Title/Abstract] OR India[Title/Abstract] OR Pakistan[Title/Abstract] OR Nepal[Title/Abstract] OR "Sri Lanka"[Title/Abstract] OR Maldives[Title/Abstract] OR China[Title/Abstract] OR Japan[Title/Abstract] OR Korea[Title/Abstract] OR "South Korea"[Title/Abstract] OR "North Korea"[Title/Abstract] OR Mongolia[Title/Abstract] OR Taiwan[Title/Abstract] OR Kazakhstan[Title/Abstract] OR Kyrgyzstan[Title/Abstract] OR Tajikistan[Title/Abstract] OR Turkmenistan[Title/Abstract] OR Uzbekistan[Title/Abstract] OR Indonesia[Title/Abstract] OR Malaysia[Title/Abstract] OR Philippines[Title/Abstract] OR Singapore[Title/Abstract] OR Thailand[Title/Abstract] OR Vietnam[Title/Abstract] OR Laos[Title/Abstract] OR Cambodia[Title/Abstract] OR Myanmar[Title/Abstract] OR Brunei[Title/Abstract] OR "Timor-Leste"[Title/Abstract]) |

a CoCoPoP = Condition (Co); Context (Co) and Population (Po)

b MeSH = Medical Subject Headings

Boolean operators (AND, OR, NOT), truncations, and filters (free full texts, keywords) will be applied to refine the search results.

## Data extraction

Data extraction will be conducted by two independent reviewers using a pre-tested and uniform Google Excel form. The reviewers will independently conduct study screening, quality assessment, and data extraction. Any disagreements will be verified through discussion and consensus in consultation with a third reviewer. The extraction form will include author and study details (study design, sampling technique, sample size, year of study, data collection method), geographic location, income level, demographic detail details (e.g., age range, gender, marital status), measures for assessing IPV, suicidal behaviors (SI, SP, SA), time frame for considering suicidal behaviors (lifetime, past year etc.). In addition, the associated factors both risk and protective factors of IPV and SB will be extracted, while mediator variables influencing the IPV–SB relationship will also be captured when available. All numerical outcomes reported in the included studies such as correlation coefficients, odds ratios, prevalence estimates, mean differences, and standardized mean differences will be extracted for meta-analysis along with their confidence intervals when available. Primary effect measures will be taken as reported from included studies. When information is incomplete (e.g., missing standard errors, confidence intervals), effect sizes will be calculated using established conversion formulas [17] or extrapolated from available statistics such as *p*-values, test statistics, or sample sizes.

## Data synthesis and statistical analysis

For quantitate synthesis, a random effect model meta-analysis (REML) will be conducted using R studio v.4.3.2 "meta" packages. As the included studies are expected to derive from diverse settings, and background with high heterogeneity a REML model will be adopted to account for both within and between study variance. Statistical heterogeneity will be assessed using the Q statistic, and its degree was estimated using the $I^2$ statistic. Range of $I^2$ heterogeneity will be interpreted as either mild ($I^2<25\%$), moderate ($50\%<I^2<25\%$), and severe or high severe ($I^2>75\%$), with REML applied for when $I^2>50\%$ [18] (Fig 3). In addition, a random-effects model (REM) using restricted maximum likelihood (REML) will be applied. Furthermore, subgroup analysis and sensitivity analysis will be conducted to explore potential sources of heterogeneity.

The included studies are expected to report associations between IPV, suicidal behaviours using different effect measures. For the meta-analysis, Pearson's correlation coefficient will be used as the common effect size. Other reported measures including standardized mean differences, odds ratios, and standardized beta coefficients will be converted to Pearson's correlation coefficients ($r_s$) using established conversion formulas [17]. To address the potential instability of variance in Pearson's r, correlation coefficients will be converted to Fisher's z-scores using the formula $z=0.5\times[\ln(1+r)-\ln(1-r)]$ [19]. All analyses will use Fisher's z as the effect size with 95% confidence intervals (CIs). Effect sizes will be interpreted as small (0.1), moderate (0.3), or large (0.5)[17]. Pooled prevalence of IPV and suicidal behaviors (SI, SA, SP) will be visualized using forest plot across the studies reporting prevalence estimates (Fig 4A). Publication bias will be evaluated using funnel plots and Egger's test when ≥10 studies will be included, if <10 studies, publication bias will not be formally tested due to low statistical power. Furthermore, sensitivity analysis will be performed using the jack-knife method to assess the robustness of the pooled estimates and determine whether any single study disproportionately influences the overall pooled estimated results [20] (Fig 4B). These analyses will be limited to associations between overall IPV

## Advanced Interpretation of I² Heterogeneity and Model Selection

Based on Cochrane & REML guidelines for meta−analysis

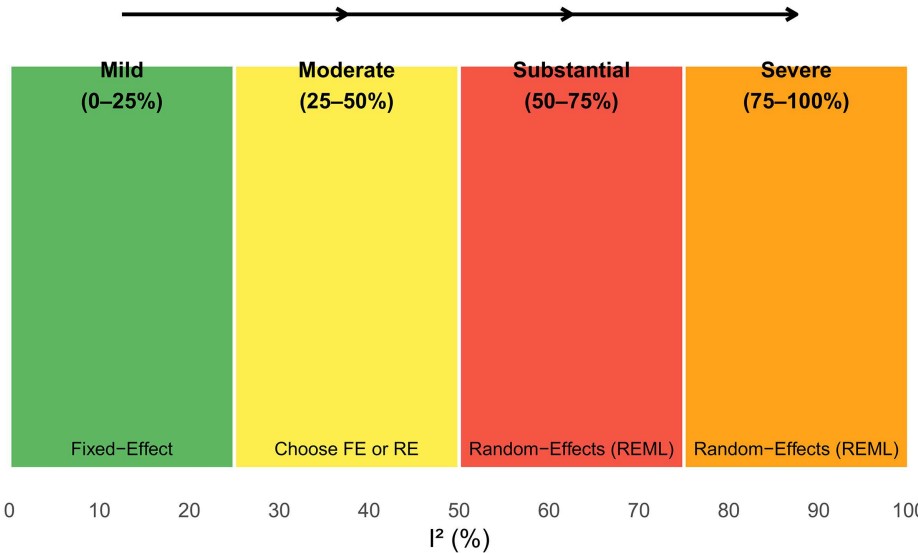

**Fig 3. Advanced Interpretation of I² Heterogeneity and Model Selection.** This figure illustrates the interpretation ranges of I² heterogeneity mild (0–25%), moderate (25–50%), substantial (50–75%), and severe (75–100%), and guides the choice between Fixed-Effect and Random-Effects (REML) models based on Cochrane recommendations.

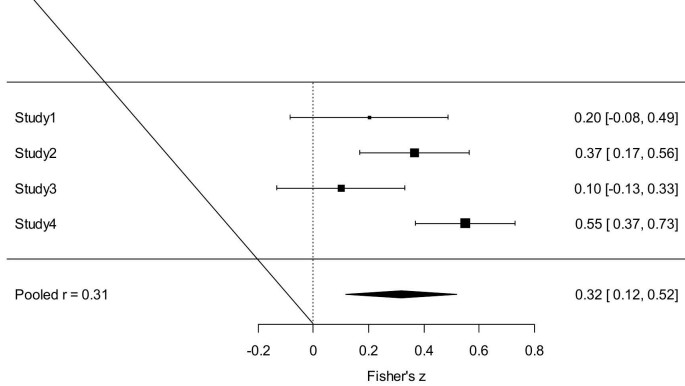

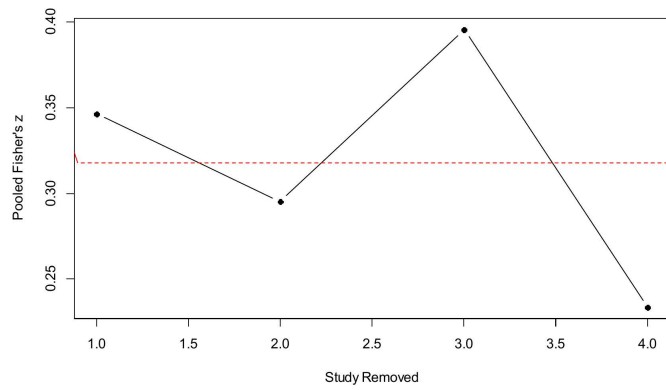

**4A: Forest plot with pooled estimates**

**4B: Jackknife Sensitivity Analysis**

**Fig 4. Forest Plot and Jackknife Sensitivity Analysis of Pooled Fisher's z Estimates.** Panel A presents the forest plot showing individual study Fisher's z values with 95% confidence intervals and the pooled correlation estimate (r = 0.xx). Panel B shows the jackknife sensitivity analysis assessing the robustness of the pooled estimate by sequentially removing each study.

and suicidal behaviors, as at least ten studies are required per group. Associated factors related to IPV and SB will be extracted and narratively synthesized due to variability in measurement across studies. Where sufficient data will be available, subgroup analyses (region, type of IPV, timeframe) will be performed to explore potential sources of heterogeneity across studies. Potential moderators of the IPV–SB relationship will also be summarized descriptively. Moderator effects will be explored via subgroup analysis (study setting, income settings, marital status, population characteristics, measurement tools, and geographic region), and meta-regression to examine potential sources of heterogeneity.

## Ethics statements

As the study is based on a review of secondary data, and thus, it does not require any formal ethical approval.

## Patient and public involvement

The study protocol does not involve direct or indirect participation of patients or members of the public.

## The status and timeline of the study

The primary screening (title and abstract) is expected to be completed by November 2025, with data extraction anticipated to be completed within February 2026. Final analysis and reporting are projected to be completed within August, 2026. Any deviations from the planned timeline or protocol will be clearly documented in the final manuscript.

## Discussion

The systematic review and meta-analysis will extract the available evidences across 48 Asian countries to investigate the prevalence, association, and related factors of IPV and SB among the women of reproductive age. Prior studies have reported that globally, one in three women experiences IPV, including physical and/or sexual violence, during their lifetime. The prevalence varies by region: 22% in high-income countries, 31% in the Eastern Mediterranean, and 33% in Africa and South-East Asia. The World Health Organization (WHO) has identified gender inequality and societal norms

that tolerate violence against women as the root causes of IPV worldwide [2]. The violence has both short- and long-term health consequences including physical, mental, sexual and reproductive health events. It has been reported women with IPV (mostly physical and sexually abused) were 1.5 times higher risk of sexually transmitted diseases (STDs) and 41% more likely to have pre-term birth as long-term consequences [2, 7]. The economic and social burden of IPV is substantial nearly the lifetime cost ~ $103,767 per female among the US adults [21]. Furthermore, studies have investigated IPV as the precursor of suicide (6.1%) particularly among the young adults globally. A study using the US National Violent Death Reporting System (2014–2018) reported that most IPV-related suicides involved male perpetrators, while younger female victims were more often affected by physical abuse and males by psychological abuse [22]. Region-specific evidence is recommended because cultural norms, reporting patterns, and support systems differ across settings, requiring IPV prevention and response strategies to be tailored to local contexts.

## Strengths and limitations

The study will aim to evaluate and integrate findings from observational cross-sectional studies following standard guidelines and corresponding checklists. While heterogeneity among included studies may pose challenges for meta-analysis and pooled estimates. Secondly, reliance on cross-sectional data may precludes establishing causality, leaving open the possibility that suicidal behavior might precede or cooccur with IPV in complex bidirectional relationships. Furthermore, underreporting of IPV or suicidal behaviors, particularly in regions with strong stigma, may have underestimated the true association.

## Ethics and dissemination of results and publication policy

This study synthesizes evidence solely from previously published available research and therefore does not require ethical approval form any review board, as no primary data will be collected. The protocol documents are included within the manuscript and as S1 Table. Further data could be requested from the authors.

Upon completion, key findings will be disseminated through national and international conferences, webinars, and other academic platforms. Additionally, tailored summaries will be shared with policymakers (government ministries and departments), clinicians (emergency medicine specialist, psychiatrists, psychologists, gynecologists), non-governmental organizations (NGOs), and other stakeholders such as law enforcement and legal services, medical universities, community and advocacy groups, international development partners, and the education ministry to inform practice, guide future research, and support intervention planning.

## Potential impact

LMICs including Asian countries had been reported higher prevalence of IPV (27%) and its sub-types including physical (23%), sexual (9.5%), and emotional violence (12.5%) [23]. Prior studies had identified women in Asia were exposed to IPV had higher risk of SB due to several socio-cultural, psychological, and mental disorders including depression, anxiety, and post-traumatic stress. However, early screening for psychological distress using culturally validated assessment tools may serve as an essential strategy for preventing SB among Asian victims of violence. In addition, examining the underlying drivers of IPV and comparing these patterns with global evidence will help identify context-specific factors, and gender norms that sustain violence and elevate mental health risks.

## Supporting information

**S1 Table. Search strings for different pre-defined databases (PubMed, Scopus, PsycINFO, Web of Science, EMBASE, CINAHL, and Google Scholar).**
(DOCX)

**S1 File. PRISMA-P Checklist 2025.**
(DOCX)

## Acknowledgments

The authors sincerely acknowledge each affiliated author's institution specially iccdr,b, Dhaka, Bangladesh, through which the databases library could be accessible.

## Author contributions

**Conceptualization:** Mantaka Rahman.

**Data curation:** Mantaka Rahman, Habiba Kabir, Anika Naowar Chowdhury, Md. Babu Raihan Mia, Tamal Saha.

**Formal analysis:** Mantaka Rahman, Anika Naowar Chowdhury.

**Investigation:** Afroza Tamanna Shimu.

**Supervision:** Ummul Khair Alam.

**Visualization:** Mantaka Rahman, Afroza Tamanna Shimu.

**Writing – original draft:** Mantaka Rahman, Habiba Kabir.

**Writing – review & editing:** Mantaka Rahman, Habiba Kabir, Anika Naowar Chowdhury, Md. Babu Raihan Mia, Tamal Saha, Afroza Tamanna Shimu, Ummul Khair Alam.

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
