## [Decision Letter · Decision Letter 0]

11 Mar 2026

Dear Dr. Rahman,

Thank you for submitting your manuscript to PLOS ONE. After careful consideration, we feel that it has merit but does not fully meet PLOS ONE’s publication criteria as it currently stands. Therefore, we invite you to submit a revised version of the manuscript that addresses the points raised during the review process.

We look forward to receiving your revised manuscript.

Kind regards,

Saeed Ahmad

Academic Editor

PLOS One

Journal Requirements:

4.If the reviewer comments include a recommendation to cite specific previously published works, please review and evaluate these publications to determine whether they are relevant and should be cited. There is no requirement to cite these works unless the editor has indicated otherwise.

Reviewers' comments:

Reviewer's Responses to Questions

**Comments to the Author**

1. Does the manuscript provide a valid rationale for the proposed study, with clearly identified and justified research questions?

Reviewer #1: Partly

2. Is the protocol technically sound and planned in a manner that will lead to a meaningful outcome and allow testing the stated hypotheses?

Reviewer #1: Yes

3. Is the methodology feasible and described in sufficient detail to allow the work to be replicable?

Reviewer #1: Yes

4. Have the authors described where all data underlying the findings will be made available when the study is complete?

The PLOS Data policy requires authors to make all data underlying the findings described in their manuscript fully available without restriction, with rare exception, at the time of publication. The data should be provided as part of the manuscript or its supporting information, or deposited to a public repository. For example, in addition to summary statistics, the data points behind means, medians and variance measures should be available. If there are restrictions on publicly sharing data—e.g. participant privacy or use of data from a third party—those must be specified.requires authors to make all data underlying the findings described in their manuscript fully available without restriction, with rare exception, at the time of publication. The data should be provided as part of the manuscript or its supporting information, or deposited to a public repository. For example, in addition to summary statistics, the data points behind means, medians and variance measures should be available. If there are restrictions on publicly sharing data—e.g. participant privacy or use of data from a third party—those must be specified.

Reviewer #1: Yes

5. Is the manuscript presented in an intelligible fashion and written in standard English?

Reviewer #1: No

You may also provide optional suggestions and comments to authors that they might find helpful in planning their study.

Reviewer #1: Very interesting title: Despite it is current issues which need interventions, it is overlooked issues.

Design and registration

• Line 151: please provide the link

Eligibility criteria

• Line 167 : This is mixed method study and please use CoCoPo for prevalence which condition context and population, and PEO for exposure review and risk review population, exposure and outcome. compactor and s- study is not needed for this particular review. for study type you can describe under inclusion and exclusion criteria please cit PRISMA 2020 guideline

Line 192: Table 1: Example Search Strategy for PubMed Database (PICOS Framework)

• This is mixed method study and please use CoCoPo for prevalence which condition context and population, and PEO for exposure review and risk review population, exposure and outcome. compactor and s- study is not needed for this particular review. for study type you can describe under inclusion and exclusion criteria please cit PRISMA 2020 guideline

Quality (risk of bias) assessment

• Line 196: RoB is not JBI tool. it is Cochran tool. so, which tool do you plan to use JBI or Cochrane?

• why only cross-sectional studies?

Data management

• line 197: Line 208: Do you mean study selection?

Data management

• Do you mean Data selection?

Data extraction

• Line 219: pretested or structured? which one do you thick better?

• Line 221: what if not agree?

Data synthesis and statistical analysis:

• Line 240: what other methods?

• Line 254: if the number of studies are >=10

• Line 2540: For what do you plan to perform sensitivity analysis

• Line 261: what about subgroup analysis and publication bias analysis?

Potential Impact

Line 318 not necessary at this stage. you will address it latter

.

Reviewer #1: No

---

## [Author Response · Author response to Decision Letter 1]

13 Mar 2026

13th March, 2026.

Dear Editor

Saeed Ahmad

Academic Editor

PLOS One

Subject: Revision submission for Manuscript (ID #PONE-D-25-62887)

Dear Editor,

Thank you for informing me the decision regarding my manuscript. I sincerely appreciate the reviewers’ time and constructive comments. After carefully reviewing the feedback, we firmly hope and believe that it will substantially improve the overall quality of the manuscript. The point by point response of reviewer comments are attached below.

We hope the revised version would be strengthen scientifically and acceptable for publication in PLOS One. Thank you very much, and we are looking forward to hearing from you soon.

On behalf of all the authors,

Dr. Mantaka Rahman (Corresponding author)

icddr,b

Dhaka, Bangladesh.

RESPONSES TO THE REVIEWER 1

1. Reviewer’s Comment: Very interesting title: Despite it is current issues which need interventions, it is overlooked issues.

Authors’ response: We thank the reviewer for his concern. The suggested point by point responses are attached below.

2. Reviewer’s Comment: Design and registration

• Line 151: please provide the link

Authors’ response: The link of the registration has been addressed accordingly (p6, Line 159).

3. Reviewer’s Comment: Eligibility criteria

• Line 167: This is mixed method study and please use CoCoPo for prevalence which condition context and population, and PEO for exposure review and risk review population, exposure and outcome. compactor and s- study is not needed for this particular review. for study type you can describe under inclusion and exclusion criteria please cit PRISMA 2020 guideline

Authors’ response: The suggested following changes have been incorporated accordingly. We have added CoCoPoP in the Table 1; incorporated PEO for the review by excluding comparator and study. Additionally, we have also cited PRISMA 2020 in Line 135 accordingly in the revised manuscript.

4. Reviewer’s Comment: Line 192: Table 1: Example Search Strategy for PubMed Database (PICOS Framework)

• This is mixed method study and please use CoCoPo for prevalence which condition context and population, and PEO for exposure review and risk review population, exposure and outcome. compactor and s- study is not needed for this particular review. for study type you can describe under inclusion and exclusion criteria please cit PRISMA 2020 guideline

Authors’ response: The suggested following changes have been incorporated accordingly in the following section.

5. Reviewer’s Comment: Quality (risk of bias) assessment

• Line 196: RoB is not JBI tool. it is Cochran tool. so, which tool do you plan to use JBI or Cochrane?

• why only cross-sectional studies?

Data management

• line 197: Line 208: Do you mean study selection?

Authors’ response: We acknowledge reviewer concern. We do plan to use JBI tool to assess the quality of the included papers and thus revise the section accordingly. In addition, only cross-sectional studies will be included as they are the most appropriate design for estimating prevalence and associated factors in the target population (Line 168-170). In the data management section, we have revised the line for better clarification (p8, Line 188).

6. Reviewer’s Comment: Data management

• Do you mean Data selection?

Authors’ response: Yes, and the section heading has been revised accordingly.

7. Reviewer’s Comment: Data extraction

• Line 219: pretested or structured? which one do you thick better?

• Line 221: what if not agree?

Authors’ response: In previous Line 219 we think pre-tested would be perfect to use and revised accordingly. We have clarified in the revised version that if the independent reviewer were not agreed, third reviewer will intervein and give a solution furthermore.

8. Reviewer’s Comment: Data synthesis and statistical analysis:

• Line 240: what other methods?

• Line 254: if the number of studies are >=10

• Line 2540: For what do you plan to perform sensitivity analysis

• Line 261: what about subgroup analysis and publication bias analysis?

Authors’ response: In previous line 240 we have added other methods like random-effect model and complemented by subgroup and sensitivity analysis to explore sources of heterogeneity. In previous line 254 we have added the reviewer suggested line accordingly. We have also clarified the reason for performing sensitivity analysis to find if any single study effect on the pooled estimated results or not. Lastly, we do add our plan to have subgroup and publication bias analysis (Funnel plot and egger’s test) in the revised manuscript accordingly (p9, Line 212-244).

9. Reviewer’s Comment: Potential Impact

Line 318 not necessary at this stage. you will address it latter

Authors’ response: The suggestion has been addressed accordingly in the revised version and the line has been removed.

Thank you for your kind consideration.

---

## [Decision Letter · Decision Letter 1]

26 Mar 2026

Prevalence, Associated factors, and Association of Intimate Partner Violence and Suicidal Behaviors Among Women of Reproductive Age in Asia: Protocol for a Systematic Review and Meta-analysis of Cross-Sectional Studies

PONE-D-25-62887R1

Dear Dr. Rahman,

We’re pleased to inform you that your manuscript has been judged scientifically suitable for publication and will be formally accepted for publication once it meets all outstanding technical requirements.

Kind regards,

Saeed Ahmad, PhD

Academic Editor

PLOS One

Additional Editor Comments (optional):

Reviewers' comments:

Reviewer's Responses to Questions

**Comments to the Author**

1. Does the manuscript provide a valid rationale for the proposed study, with clearly identified and justified research questions?

Reviewer #1: Yes

2. Is the protocol technically sound and planned in a manner that will lead to a meaningful outcome and allow testing the stated hypotheses?

Reviewer #1: Yes

3. Is the methodology feasible and described in sufficient detail to allow the work to be replicable?

Reviewer #1: Yes

4. Have the authors described where all data underlying the findings will be made available when the study is complete?

The PLOS Data policy requires authors to make all data underlying the findings described in their manuscript fully available without restriction, with rare exception, at the time of publication. The data should be provided as part of the manuscript or its supporting information, or deposited to a public repository. For example, in addition to summary statistics, the data points behind means, medians and variance measures should be available. If there are restrictions on publicly sharing data—e.g. participant privacy or use of data from a third party—those must be specified.requires authors to make all data underlying the findings described in their manuscript fully available without restriction, with rare exception, at the time of publication. The data should be provided as part of the manuscript or its supporting information, or deposited to a public repository. For example, in addition to summary statistics, the data points behind means, medians and variance measures should be available. If there are restrictions on publicly sharing data—e.g. participant privacy or use of data from a third party—those must be specified.

Reviewer #1: Yes

5. Is the manuscript presented in an intelligible fashion and written in standard English?

Reviewer #1: Yes

You may also provide optional suggestions and comments to authors that they might find helpful in planning their study.

Reviewer #1: Great response. The authors address all comments.is can be accepted for publications in PLOS one Journal

.

Reviewer #1: No

---

## [Editor Report · Acceptance letter]

PONE-D-25-62887R1

PLOS One

Dear Dr. Rahman,

I'm pleased to inform you that your manuscript has been deemed suitable for publication in PLOS One. Congratulations! Your manuscript is now being handed over to our production team.

Kind regards,

on behalf of

Dr. Saeed Ahmad

Academic Editor

PLOS One